# Understanding Cystic Fibrosis Comorbidities and Their Impact on Nutritional Management

**DOI:** 10.3390/nu14051028

**Published:** 2022-02-28

**Authors:** Dhiren Patel, Albert Shan, Stacy Mathews, Meghana Sathe

**Affiliations:** 1Division of Pediatric Gastroenterology, Hepatology and Nutrition, School of Medicine, Saint Louis University, St. Louis, MO 63104, USA; 2Department of Pediatric Gastroenterology, Hepatology and Nutrition, School of Medicine, Saint Louis University, St. Louis, MO 63104, USA; albert.shan@health.slu.edu (A.S.); stacy.mathews@health.slu.edu (S.M.); 3Division of Pediatric Gastroenterology and Nutrition, University of Texas Southwestern, Children’s Medical Center Dallas, Dallas, TX 75235, USA; meghana.sathe@utsouthwestern.edu

**Keywords:** cystic fibrosis, nutrition, comorbidities, multidisciplinary care, patient outcomes, growth and development, feeding difficulties

## Abstract

Cystic fibrosis (CF) is a chronic, multisystem disease with multiple comorbidities that can significantly affect nutrition and quality of life. Maintaining nutritional adequacy can be challenging in people with cystic fibrosis and has been directly associated with suboptimal clinical outcomes. Comorbidities of CF can result in significantly decreased nutritional intake and intestinal absorption, as well as increased metabolic demands. It is crucial to utilize a multidisciplinary team with expertise in CF to optimize growth and nutrition, where patients with CF and their loved ones are placed in the center of the care model. Additionally, with the advent of highly effective modulators (HEMs), CF providers have begun to identify previously unrecognized nutritional issues, such as obesity. Here, we will review and summarize commonly encountered comorbidities and their nutritional impact on this unique population.

## 1. Introduction

Cystic fibrosis (CF) is a progressive, autosomal recessive genetic disorder with a mean prevalence of 0.74 in 10,000 persons in the US. Mutations in the cystic fibrosis transmembrane conductance regulator (CFTR) protein gene lead to disruptions in the transport of chloride out of cells. This, coupled with overactive epithelial sodium channels, leads to thickened mucus secretions throughout the body, including lungs, pancreas, liver, gallbladder, and intestines [1]. The above contributes to multisystemic comorbidities (Table 1) and makes maintaining adequate nutrition status challenging in CF patients by decreasing nutritional intake, increasing metabolic demands, and decreasing intestinal absorption [2]. It is crucial to utilize a multidisciplinary team with expertise in CF to optimize growth and nutrition, given that undernourished persons with CF are more likely to have poorer clinical outcomes, such as reduced pulmonary function [2,3]. Here, we will review commonly encountered comorbidities and their nutritional impact on this unique population.

## 2. Sinusitis

Cystic fibrosis chronic rhinosinusitis (CFCRS), a common finding in people with cystic fibrosis, is defined by at least 12 weeks of persistent sinus inflammation with signs and symptoms of sinusitis. In CF, there is a blunted cell membrane transport of chloride ions, leading to decreased amounts of water that crosses into mucosal secretions. This results in thick, inspissated mucus and poor mucociliary clearance, which can give rise to secondary bacterial colonization and recurrent sinopulmonary infections. Long-term mucosal infection and inflammation predispose patients with CF to chronic rhinosinusitis [4].

While some individuals may be asymptomatic, affected persons may present with headache, facial pain or pressure, anosmia or hyposmia, chronic nasal congestion, nasal discharge, polyps, or mucosal edema. In infants, as obligate nose breathers, nasal obstruction and congestion may affect the ability to feed [5]. Olfaction dysfunction may lead to poor appetite, food aversion, and subsequently suboptimal nutritional intake. In addition, the literature suggests that paranasal sinuses can harbor bacteria that lead to pulmonary exacerbations and further worsening of nutritional status, as discussed in the following section [6].

Treatments can include topical steroids, nasal saline irrigation, and mucolytic agents, such as dornase-alpha [7,8]. There are varied data supporting the use of antibiotics and endoscopic surgery [9]. There is also research supporting the role of CFTR protein modulation in treating CF-related sinus disease. Ivacaftor, a highly effective CFTR modulator therapy primarily utilized for gating mutations [10], was reported to improve CF-related chronic rhinosinusitis (CF CRC) symptoms in persons with medically and surgically intractable Cf CRC and has shown to improve food intake and appetite [11,12].

## 3. Lung Disease

There are significant data from the Cystic Fibrosis Foundation Patient Registry (CFFPR) regarding the correlation between pulmonary function, resting energy expenditure, and nutritional status [13]. Below, we will discuss literature highlighting the relationship between lung function and nutrition.

People with malnutrition have significantly lower mean vital capacity and forced expiratory volumes (FEV). One study found that adolescents who were malnourished presented with significant declines in pulmonary function, as measured by forced expiratory volume, compared to their peers. In one year, individuals who lost more than 5% of their weight had FEV values that were 16.5% lower than predicted. On the other hand, participants who gained weight had an increase in FEV that was 2.1% higher than predicted [14]. Another study found that at higher weights and with developmentally appropriate weight gain, children exhibit better improvements in average forced expiratory volume, which can be used as a surrogate marker of lung health [15]. In a study by Zemel et al., it was found that nutritional status was positively predictive of improvements in pulmonary function in children with mild pulmonary disease [13]. It is possible that deterioration of lung function leads to worsening nutritional status by way of decreasing intake and increasing nutritional requirements. An alternative is that deteriorating nutritional status may lead to poor pulmonary function by weakening respiratory musculature. Children with worsening pulmonary function test results should be evaluated for nutritional deficits, as it can be indicative of worsening disease.

Sino-pulmonary symptoms of CF can include persistent cough and nasal drainage. These can impact a child’s ability to consume calories due to blunted olfaction and taste. It may also impact the process of eating when cough is persistent. Additionally, chronic bacterial infections, such as *Pseudomonas aeruginosa,* can lead to chronic inflammation. Chronic inflammation can be associated with adiposity, as well as reductions in linear growth [16]. Metabolic overload and adipocyte enlargement can trigger apoptosis and a subsequent inflammatory reaction. Additionally, research shows that excessive fat ingestion, particularly without concomitant antioxidant ingestion, may contribute to inflammation linked with obesity [17]. Welsh et al. suggests that increased adiposity leads to higher levels of C-reactive protein, a common marker of inflammation [18]. In a similar vein, weight loss in otherwise healthy obese patients is associated with decreases in C-reactive protein levels [19]. Further studies need to be conducted to better understand the relationship between obesity and inflammation in patients with CF.

## 4. Gastroesophageal Reflux Disease

Gastroesophageal reflux disease (GERD) is defined as troublesome symptoms and/or complications occurring when gastric contents pass into the esophagus [20]. It is significantly more common in children with CF [21], and data have suggested that infants with CF are four times more likely to have GERD than infants without CF [22]. GERD can be one of the most common and challenging-to-treat symptoms in patients with CF, with some estimates stating up to 80% of this population reporting GERD [23,24].

The development of GERD is multifactorial, including lower esophageal sphincter dysfunction and increased transient lower esophageal sphincter relaxations (TLESRs) [23,25,26]. In CF, other factors include increased negative thoracic inspiratory pressure from pulmonary disease, increased intra-abdominal pressure from coughing, poor clearance of esophageal digestive contents, gastric dysmotility [27], or CF treatment side effects, including a high-fat diet, gastrostomy feeds, medication side effects, and physiotherapy [25].

The diagnosis of GERD can be made clinically in a person who exhibits typical symptoms, including recurrent emesis or regurgitation, heartburn, abdominal or chest pain, respiratory symptoms, and poor weight gain, most marked in infants [28,29]. Barium swallowing study helps evaluate malrotation and other anatomic abnormalities. Upper gastrointestinal endoscopy may be valuable to exclude other causes or evaluate complications. While pH-metry itself has limited diagnostic utility, pH-Impedance (pH-MII) may help provide better symptom correlation with reflux events, especially when combined with high-resolution manometry (HRM). This could determine the role of reflux and aspiration in persons with diminished lung function [30,31].

The potential that GERD has in limiting caloric intake, including frequent emesis and challenges when introducing a high-fat diet, puts people with moderate-to-severe GERD at risk of malnutrition, impacting the quality of life and worsening disease severity. Currently, the association between GERD and poor pulmonary function is unclear [25]. Frequent proton pump inhibitors (PPI) use in patients with CF was associated with lower hemoglobin levels, possibly due to elevated gastric pH and decreased iron absorption [32].

There are no specific guidelines to manage GERD in patients with CF; treatment strategies, in general, include non-pharmacological, pharmacological treatment, and surgery [20]. In infants, thickened feedings, a trial of extensive hydrolyzed or amino-acid formula, and smaller volume feed at an increased frequency may help alleviate the symptom [20]. Head elevation or left lateral positioning may help treat older children. Avoidance of spicy, fried, and acidic food, as well as smaller but more frequent meals, may help symptom control. Weight control, especially in the era of HEM therapy, is helpful if the person is obese. Commonly used pharmacological therapy, such as acid suppression, can be beneficial in reflux-related erosive esophagitis; however, these medications should be used cautiously, since utility in reducing the number of reflux episodes is unclear [33]. The facts on the effect of gastric acid suppression on pulmonary function are conflicting [34,35], with some data suggesting that acid suppression results in earlier and more frequent exacerbations [34], and other suggesting that acid suppression does not affect pulmonary function [35]. Baclofen and bethanechol may be used off label before consideration of surgical fundoplication, a common invasive anti-reflux surgical procedure [20]. The benefit of fundoplication is controversial but may be indicated in refractory cases. A retrospective study reported by Boesch et al. found no changes in pulmonary function or nutrition one year post-fundoplication [36]. In contrast, Shahid et al. found that in people with uncontrolled GERD and worsening lung function, Nissen fundoplication led to significant improvement in weight, fewer pulmonary exacerbations, and slowed the decline of lung function [37,38]. In lung transplant patients, fundoplication may reduce the risk of bronchiolitis obliterans [39]. Complications from fundoplication and GERD recurrence do happen, and consideration of the potential risks and benefits of the surgery need to be evaluated for each individual.

## 5. Gastroparesis

Gastroparesis (GP) is defined by constellation of various upper GI symptoms along with delayed gastric emptying in the absence of mechanical obstruction to the passage of content from the stomach to duodenum. Normal gastric emptying requires coordination between intrinsic neurons and extrinsic input from the central and autonomic nervous systems. Balanced excitatory and inhibitory signals and appropriated transmission lead to a series of motor events, including proximal stomach accommodation, antral contractions, pyloric sphincter relaxation, and antropyloric–duodenal coordination.

There are no data on the prevalence of the pediatric population so far. At-risk populations include malnutrition, eating disorders, functional GI disorders, gastrointestinal anatomical abnormalities, connective tissue disorders, postinfectious and chronic inflammatory processes [40]. The incidence of delayed or rapid gastric emptying within the CF population has not been consistent in the literature. In a retrospective cohort study, 9 out of 239 patients enrolled had CF, and 4 (44%) of the 9 CF patients had GP [41]. A systemic review conducted by Corral, J.E. et al. showed that patients with CF have a high frequency of GP of up to 38%, with a higher prevalence for patients older than 18 years, while younger patients with untreated pancreatic insufficiency had rapid gastric emptying [42].

While the pathophysiology of GP remains unclear, various mechanisms have been proposed, including altered gut hormones in response to meals, impaired ileal brake secondary to fat malabsorption, and abnormalities in the enteric nervous system [43,44,45]. Frequent use of opiates and anticholinergics can also decrease gastric emptying and intestinal transit time. Diabetes does not seem to play a significant role in pediatric GP in CF [42].

In general, vomiting was the most frequent presenting symptom, followed by abdominal pain, nausea, weight loss, early satiety, and bloating [41]. Infants and younger children are more likely to experience vomiting, while adolescents predominantly manifest symptoms of nausea and abdominal pain [46]. The diagnosis of GP is often challenging in pediatric patients, given the lack of normative data and poor standardization of testing in this population [40].

GP compromises patients’ quality of life, increases financial burden, and significantly worsens nutrition status [47,48]. It often leads to food aversion, causing oral caloric intake restriction [49]. Adult studies have shown that patients with GP often have a diet deficient in calories, fat, protein, and several vitamins and minerals, on assessment [50]. A large proportion of patients consumed less than 60% of their daily energy requirements [50]. Wassem S. et al. reported a pediatric retrospective cohort of 239 patients with GP, with 27% of them experiencing weight loss [41]. While no study has focused explicitly on CF patients, there is a concern that GP will interfere with patients’ adherence to supplements and oral medication, specifically pancreatic enzyme replacement therapy (PERT), which further worsens malabsorption and leads to nutritional deficiencies. As discussed in the GERD section, frequent usage of proton pump inhibitors also puts patients at greater risk of iron deficiency and small bowel bacterial overgrowth [51].

No specific pediatric treatment guidelines have been established for GP. GP-associated nutritional deficiency could arise due to inability to take in adequate intake or due to interventions required to treat GP, such as small feeds, tube feeding, and adverse effects of medications used for GP. Diet intervention, traditionally, may start with smaller and more frequent meals; however, it raises concern for possible insufficient caloric intake in the CF population, especially in the younger children [31]. A few studies in infants reported that feeding with medium-chain triglycerides and whey–hydrolysate formula improves gastric emptying [52,53,54]. Prokinetic agents are often required. Macrolide antibiotics, such as erythromycin and azithromycin, are widely used [40,55]. Amoxicillin/clavulanate, though mainly used for small bowel bacterial overgrowth, also seems to increase gastrointestinal motility [56]. Metoclopramide has a block-box warning by the FDA due to the risk of tardive dyskinesia, which has limited its use; however, it is often still utilized after partnering with patients and families regarding potential risks due to its effectiveness. Cisapride and domperidone, though effective, are not readily available in the United States due to increased risk of cardiac arrhythmias and death [57]. Other invasive treatment options for medically refractory GP, including pyloroplasty or gastric electrical stimulation, are beyond the scope of this paper and can be reviewed elsewhere [40]. Lastly, although with risks of complications, including food aversion, tube feeding (slow nasogastric or nasojejunal/gastrojejunal feeding) may be indicated to optimize patients’ growth in all age groups [40].

## 6. Small Bowel Bacterial Overgrowth

Small intestine bacterial overgrowth (SIBO) has been reported in up to 40% of patients with CF [58,59]. Patients with CF have increased acidity of intestinal and biliary tract secretions due to decreased bicarbonate secretion by an injured exocrine pancreas and dysregulation of salts in biliary system due to the presence of dysfunction CFTR in cholangiocytes. Additionally, there is a slowing of intestinal motility, obstruction from inspissated intestinal secretions, and reduced gastric acid production, all of which can be associated with the development of SIBO. Another thought is that SIBO in CF is secondary to swallowed bacteria, which is higher in individuals with CF [60].

Symptoms of SIBO can overlap with gastrointestinal manifestations of CF and typically include abdominal discomfort, flatulence, and bloating [60]. Many symptoms of SIBO can be difficult to differentiate from abdominal symptoms in CF, so SIBO should be considered when abdominal symptoms persist after other diagnoses are ruled out. Gold-standard diagnosis of SIBO is with duodenal aspiration culture, although this is rarely performed due to its invasiveness. SIBO can also be diagnosed in the general population using a lactulose hydrogen breath test. However, as this test is not thought to be reliable in CF, diagnosis is most often made clinically with symptoms of gas, bloating, and/or increased steatorrhea in the setting of optimized PERT [59].

SIBO causes malnutrition through various pathogenesis, such as bacterial injury to gut epithelium, increased bacterial consumption of host nutrients, and decreased food intake because of abdominal discomfort [61,62]. Bacteria can directly affect gut epithelia and is associated with villous blunting and epithelial inflammation. It leads to decreased fatty acid production, increased gut permeability, and carbohydrate malabsorption due to reduced absorptive surface area, bacterial sugar degradation, and impaired brush-border enzyme activity [61,63,64]. Additionally, bacterial toxin production can directly impair carbohydrate and protein absorption.

Bile acid deconjugation by intraluminal bacterial contributes to inadequate micelle formation and enterocyte injury, resulting in steatorrhea and further micronutrient deficiency. In the general population, deficiencies of lipid-soluble vitamins, iron, thiamine, nicotinamide, and vitamin B12 have been found in patients with SIBO, though this has not yet been assessed specifically in children with CF [65]. Vitamin B12 deficiency may be secondary to bacterial B12 consumption and direct inhibition of normal B12 absorption. Bacteria may compete with the host consuming macronutrient and micronutrient consumption, leading to fewer nutrients for absorption. Villous blunting secondary to inflammation may also cause carbohydrate malabsorption [61].

Treatment goals include reduction in bacteria and amelioration of nutritional deficiencies. Antimicrobials are the first-line treatment [1] and can include metronidazole and rifaximin [61,66,67]. Amoxicillin/clavulanate, which can play a role in treating CF pulmonary exacerbations, also treats SIBO by stimulating contraction of the duodenum and increasing gut motility. One study showed that oral antibiotic therapy improves fat absorption and digestion in patients with SIBO [68]. Laxatives such as polyethylene glycol are another mainstay of SIBO treatment that works by increasing the excretion of bacteria [25,59]. One study found that inhaled ipratropium, a medication often used to treat pulmonary symptoms of CF, was protective against SIBO, though its anticholinergic effects on the GI tract are thought to favor SIBO through gut stasis [60].

## 7. Constipation and Distal Intestinal Obstruction Syndrome

Constipation is a significant chronic issue in CF patients, with a prevalence of approximately 47% [69]. Distal intestinal obstruction syndrome (DIOS) is an acute partial or complete obstruction with the potential for surgical intervention, with a varying incidence that ranges from 2.3 to 11.3 per 1000 patients per year [70]. The European Society for Paediatric Gastroenterology Hepatology and Nutrition (ESPHGAN) Cystic Fibrosis Working Group has defined constipation as a gradual fecal impaction presented with either abdominal pain/distension, decreased bowel movements, or increased fecal consistency, which all resolved after using laxatives. DIOS was defined as acute abdominal pain or distension complicated with signs of complete or incomplete ileocecum fecal obstruction [71].

Constipation and DIOS share some common pathophysiology. They may both result from the CFTR protein dysfunction, which may alter the intestinal fluid composition and induce thick mucus in the small bowel, leading to the anomalous intestinal milieu, biofilm formation, and dysbiosis.

Both are associated with a history of meconium ileus (MI) and inadequate fluid intake [69,72]. Though dysmotility is considered to be a potential cause for DIOS, objective motility assessments are lacking [31]. Additional risk factors for DIOS include more severe CF genotypes, history of DIOS, pancreatic insufficiency, and cystic-fibrosis-related diabetes (CFRD) [72]. There has been an inconsistent association between PERT with constipation and DIOS development [73,74].

Even with well-characterized clinical symptoms by the proposed criteria, the diagnosis remains challenging and requires careful history, physical exam, and possible imaging studies. Other differentials, such as SIBO, should be considered as significant symptom overlap.

While it is concerning that patients with chronic or recurrent abdominal symptoms have suboptimal nutrition intake, limited studies have described no association between constipation/DIOS and malnutrition. Lavie et al. performed a retrospective multicenter review on 350 patients with 20 years of follow-up; there was no statistical difference of Body Mass Index (BMI)/weight between the DIOS and control (non-DIOS) groups [75]. Mentessidou et al. conducted a 10-year retrospective review on 53 CF patients who operated on MI and found no growth impairment; however, the loss of follow-up may have confounded the result. A prospective cross-sectional study on 105 CF children by Stefano et al. also showed no statistical difference in malnutrition between the constipated and non-constipated groups [76]. Studies on the relationship between nutritional outcome and meconium ileus, an intraluminal obstruction over the distal ileum and ileocecal valve at birth caused by viscid meconium, have shown contradictory results. A further systemic review may help better characterize the association [77,78,79].

Although there is no clear evidence, exercise and appropriate fluid and fiber intake are recommended. Inadequate PERT, including poor adherence and under-dosing, is unlikely to play a role in DIOS but should be assessed as part of the overall dietetic review in CF. Proactive conservative treatment with laxatives, such as polyethylene glycol, is essential.

## 8. Other Gastrointestinal Enteropathies

With appropriate nutritional intervention, patients with refractory gastrointestinal symptoms or persistent malnutrition should be referred to a gastrointestinal specialist for further investigation, since other disorders may coexist with CF that can heavily affect nutrition.

Celiac disease is a gluten-induced T-cell-mediated systemic autoimmune disease in genetically predisposed populations. Celiac disease and CF share similar clinical presentations, including diarrhea, malnutrition, abdominal distension, abdominal pain, and liver involvement. Besides being two different entities, they do coexist. The true incidence of the coexistence is unclear, but current evidence suggests an up to three-fold increase in celiac disease in CF patients compared to those without CF. The prevalence of both celiac disease and CF was calculated as 1 in 2,000,000–1 in 5,900,000 [80]. A higher antigen load from increased intestinal permeability, inflammation, and pancreatic exocrine insufficiency secondary to CFTR dysfunction might be the leading cause [81]. Therefore, while a gluten-free diet is the standard treatment, ivacaftor has been proposed as a possible treatment strategy. Interestingly, there are cases found to have celiac disease while on ivacaftor [82]. Regardless, the clinician should have a high index of suspicion when patients with CF are not improving as expected with optimal nutrition supplementation.

Whether there is an increased prevalence of coexisting inflammatory bowel disease (IBD) remains controversial. Ongoing and persistent GI symptoms in these cases significantly contribute to decreasing ability to take adequate oral intake, as well as increased energy losses due to malabsorption. Additionally, gluten-free diets add another layer of barrier to food selection that could further restrict patients’ caloric intake. Patients with CF and enteropathies also have increased metabolic demand, which contributes to persistence of malnutrition.

## 9. Functional Gastrointestinal Disorders

Patients with CF have chronic abdominal pain, with 60% of children and 36% of adults reported in a comparative study [83]. Another pilot prospective evaluation documented around 6% of the study population has recurrent abdominal pain [84]. Visceral hypersensitivity causing abdominal pain may be a plausible etiology in CF, given the presence of CFTR in neurons [85], and functional abdominal pain disorders (FAP) should be considered if the symptoms cannot be fully explained by another medical condition after appropriate evaluation. Currently, the true prevalence of FAP is unknown in patients with CF.

According to ROME IV criteria, functional gastrointestinal disorders (FGIDs) include functional nausea and vomiting disorders, functional abdominal pain disorders (FAP), and functional defecation disorders. FAP consists of functional dyspepsia, irritable bowel syndrome, abdominal migraine, and functional abdominal pain not otherwise specified. Therefore, an individual’s symptoms vary and can include chest pain, heartburn, dysphagia, dyspepsia, epigastric or other abdominal pain, emesis, and changes in stool frequency or consistency. Chronic discomfort may cause poor appetite that compromises a child’s nutritional status through missed meals and decreased food intake.

Pharmacological interventions include antispasmodics and mood stabilizers. Nonpharmacological therapies consist of hypnosis, cognitive-behavioral therapy, and other complementary therapies, such as mindfulness, and certain herbs, such as peppermint, chamomile, and caraway [86,87]. It is critical to emphasize the support from psychology in the care of these patients.

## 10. Disordered Eating

In the management of cystic fibrosis, there is a strong emphasis on high caloric and fat diet and maintaining or gaining weight. It is unclear how these may be associated with the development of eating disorders, such as anorexia nervosa and avoidant restrictive food intake disorder (ARFID). Consequences of CF, such as delayed puberty, low body weight, and emphasis on food management, may put these patients at risk of developing disordered eating habits. Additionally, psychosocial factors, including the demands of having a chronic illness, social isolation, and low self-esteem, may further increase risk of developing eating disorders [88].

In adults with CF, it has been reported that concerns about body image increases with declining nutritional status [89]. Disordered eating behaviors do not necessarily meet the criteria for an eating disorder, hence early recognition can help identify patients at risk. An eating disorder screening tool has been developed to identify disordered eating behaviors in patients with CF, though its validity has yet to be established [90]. In the CF population, there is an emphasis on maintaining optimal weight. Adolescent patients with chronic illness affecting diet and emphasis on maintaining weight have an increased risk of developing disordered eating behavior compared to their peers [91]. CF patients are no different when it comes to the risk of disordered eating. In fact, it has been shown that adolescents with CF have similar rates of eating disorders and self-esteem as their peers who do not have CF [90].

Some studies have shown that megestrol acetate, cyproheptadine, dronabinol, and mirtazapine may be used as appetite stimulants in patients with CF and anorexia nervosa [92]. It is important to consider eating disorders in the differential diagnosis in patients with CF who have unexplained weight loss and have risk factors for developing disordered eating for early identification and management.

## 11. Cystic-Fibrosis-Related Liver Disease

Cystic-fibrosis-related liver disease (CFLD), as an independent risk factor with a mortality rate of 2.5%, is the third leading cause of death in CF [93,94,95]. The term covers a full spectrum of diseases, including neonatal cholestasis, elevated liver enzymes, imaging abnormalities, histological changes, and gallbladder and biliary tract disorders. Severe CFLD is defined based on the presence of cirrhosis and portal hypertension [96]. The prevalence of CFLD ranges from <5% to 68% depending on the diagnostic criteria used [97,98]. The incidence of severe CFLD is 5–10% in the pediatric community [99,100]. Most of these arose in the first decade of life and developed with related complications during the second decade [95].

The pathogenesis of CFLD is not fully understood but seems to be multifactorial, including abnormal cholangiocyte function and altered biliary secretion, abnormal protein–protein interactions, abnormal innate inflammatory response, gut dysbiosis, genetics, and most recently proposed, obliterative portal venopathy [101,102]. Risk factors for CFLD include male sex, the presence of two severe mutations (loss-of-function CFTR), presence of SERPINA 1Z allele, exocrine pancreatic insufficiency, and CFRD [95,103].

Most patients are asymptomatic, and the diagnosis is challenging given the lack of consensus. ESPGHAN has proposed a diagnostic criterion with at least two of the following variables: (1) hepatomegaly or splenomegaly; (2) persistently elevated liver enzymes without other causes identified (>12 months, at least three consecutive labs); (3) abnormal liver ultrasound findings; and (4) abnormal liver histopathology, if indicated [96]. Studies have investigated numerous non-invasive biomarkers and imaging, hoping to predict or assess the severity of pediatric CFLD, such as GGT, aspartate aminotransferase to platelet ratio index (APRI), fibrosis-4 (Fib-4), liver ultrasound, transient elastography, MR elastography, and supersonic shear-wave elastography (SSWE) [95,104,105,106,107].

Patients with CFLD, especially those with a severe phenotype, are at risk of significant malnutrition, micronutrient deficiencies and worse lung function [108]. The underlying causes are complex, including but not limited to poor intake associated with anorexia, delayed gastric emptying, and early satiety from abdominal distension (ascites, organomegaly); increased energy expenditure; and maldigestion/malabsorption [109]. With unpredictable response with CFTR modulator therapy in CFRD, although not absolutely contraindicated, the hepatotoxicity potential may limit its usage in this patient population. [110,111].

Though guidelines and position papers were published, nutrition management should involve an experienced CF dietitian in the team [96,109]. In summary, patients with CFLD need higher daily calories up to 130 ~ 150% of the estimated requirement for age, preferably achieved by increasing the fat’s proportion to 40–50% considering the risk of CFRD, with supplementation in medium-chain triglycerides and polyunsaturated fatty acids. It is essential to provide supplements to ensure adequate daily protein intake (3 g/kg/day), pancreatic enzymes to optimize fat absorption, and micronutrients, including fat-soluble vitamins and zinc, to prevent deficiency. Salt supplementation is common in CF patients but should be avoided for those with cirrhosis and portal hypertension due to the risk of developing or worsening ascites. Frequent monitoring of nutritional status is critical to evaluate the response and prevent vitamin toxicity. Enteral tube feeding is safe and may be indicated if the patient cannot achieve adequate intake.

## 12. Cystic-Fibrosis-Related Diabetes

CFRD is a common comorbidity in patients with CF and can be found in 19% of adolescents [112]. CFRD is not commonly observed in the prepubertal population, but 2% of children with CF have this diagnosis. Glucose abnormalities may be found in patients of all ages, including children as young as 3 months [113], and can reflect an increased risk of developing CFRD. Even young children with normal glucose levels had lower insulin secretion than controls after the age of two [113].

The etiology of CFRD is thought to be at least in part secondary to thick secretions leading to pancreatic obstruction, as well as due to progressive fibrosis and fatty infiltration resulting in islet cell damage [114] and pancreatic exocrine dysfunction [115]. Recent studies have found cytokine IL1-beta, which is thought to cause beta-cell apoptosis leading to insulin insufficiency in both type 1 and type 2 diabetes mellitus, in the islet cells of people with CFRD. Insulin insufficiency can lead to increased protein catabolism, which can result in malnutrition and weight loss. Early identification of CFRD is important, and the diagnostic method is similar to that of type 1 and type 2 diabetes mellitus (DM), except for HbA1C, which can be falsely low in patients with CF [116]. Screening for CFRD is recommended annually, starting at age 10 for all patients with CF, with a 2 hour oral glucose tolerance test [117].

Uncontrolled CFRD and CF-related pre-diabetes can worsen pulmonary function and have negative nutritional impacts. Studies have shown that individuals with CFRD have lower height and weight percentiles compared to their peers without diabetes [118]. A decline in growth velocity and weight can precede a diagnosis of CFRD [119]. Children with CFRD who have not yet reached peak height had lower BMI percentiles than their peers up to 2 years prior to diabetes diagnosis [120]. This phenomenon was thought to be secondary to insulin-insufficiency-triggered catabolism rather than hyperglycemia itself [121]. However, in patients with CF who do not have diabetes, weight loss can be associated with higher peak glucose levels and a higher proportion of time spent with elevated serum glucose levels [119].

Optimal glycemic control is critical in the management of CFRD, and management may differ from that for patients with DM. This is due to different nutritional needs, as well as a lower risk of cardiovascular disease. Rather than strict carbohydrate restriction recommended in general DM care, people with CFRD have higher calorie, protein, fat, and salt needs and may thus instead require careful carbohydrate counting [112,122]. In conjunction with nutritional therapies, insulin therapy is the first-line treatment to insulin insufficiency in people with CFRD and is thought to not only improve nutritional status but also lung function [117]. Oral diabetes agents, such as insulin secretagogues, metformin, and thiazolidinediones are generally not recommended in CFRD [123], though some early research suggests the insulin secretagogue repaglinide may have some utility in the treatment of early CFRD [124]. Furthermore, oral medications may be associated with gastrointestinal side effects (metformin and incretin mimetic agents) and decreased bone mineral density (thiazolidinediones) [125]. Lastly, though the role of CFTR in the beta cell is unknown, some research has shown improved insulin response to glucose with CFTR modulator therapy [126].

## 13. Renal Stone

Urolithiasis is common in adults but can also be seen in children with CF. Literature suggests that impaired fat absorption and resultant enteric hyperoxaluria may play a role in stone disease. Supersaturation of calcium and oxalate in the urine can lead to crystallization and stone formation [127]. Additionally, defects in renal calcium metabolism can lead to hypocalciuria and nephrocalcinosis [128]. One study found that enzyme supplementation was negatively associated with the presence of hyperoxaluria, though the amount of enzyme given did not seem to affect this [129].

The abdominal pain associated with urolithiasis can result in decreased food intake and appetite, further affecting nutritional status. Treatment of nephrolithiasis generally consists of hydration and analgesia. Long-term therapy can include low-oxalate diet, optimal PERT, pyridoxine supplementation, and adequate hydration [127].

## 14. Micronutrient Deficiency

Patients with CF have blunted fat absorption, leading to fat-soluble vitamin deficiencies even in patients with adequate pancreatic enzyme replacement. It is recommended that these vitamin levels are checked at diagnosis and annually, followed by appropriate supplementation [130].

### 14.1. Vitamin A

Vitamin A is essential for immunity and epithelial cell function. Studies have shown that up to 40% of patients with CF have Vitamin A deficiency [131]. It is crucial to remember that Vitamin A is an acute-phase reactant, so levels measured during acute illness may inaccurately suggest hypovitaminosis A [132]. Serum retinol is a commonly used marker of vitamin A status, though there are no universally available standard definitions for vitamin A deficiency. Other markers can include retinol-binding protein, as well as serum retinyl esters. A retinol-binding protein to retinol ratio of less than 0.8 may indicate vitamin A deficiency [133].

### 14.2. Vitamin D and Calcium

Vitamin D is important because it aids in calcium absorption and is critical for bone health. Contributors to vitamin D and calcium deficiency can include low intake, gastrointestinal malabsorption, and increased fecal calcium loss. Compared to the general population, children with CF have a higher prevalence of osteopenia and an increased risk of fractures [134,135]. It is recommended that calcium be assessed routinely. Increasing dietary consumption of calcium-rich foods can include dairy products, such as milk and cheese. Intervening early with calcium and vitamin D supplementation may address these risks.

### 14.3. Vitamin E

Vitamin E deficiency is a frequent finding in CF patients [136]. A compound of vitamin E, alpha–tocopherol, helps protect the body against oxidative damage. Vitamin E is best measured through a vitamin E to total cholesterol ratio. Vitamin E deficiency can result in neuromuscular degeneration, cognitive deficits, and eye problems. A child’s vitamin E requirement may increase with higher levels of oxidative stress during a pulmonary CF exacerbation and with chronic respiratory infections.

### 14.4. Iron

Children with CF are at risk of iron deficiency. Multiple contributing factors include malabsorption, chronic inflammation or blood loss, and inadequate intake. Notably, ferritin, which is frequently used as a marker of iron status, is an acute phase reactant and thus may be falsely elevated in inflammatory states. Serum transferrin receptors are a good marker of iron stores that are not affected by inflammatory states; however, it may be difficult to test in a clinical setting. A simple marker can include checking hemoglobin and hematocrit annually or an iron panel containing ferritin and total iron-binding capacity [3].

### 14.5. Zinc

Zinc deficiency is common in patients with cystic fibrosis, and PERT can assist with zinc absorption. Symptoms of low zinc can include immunocompromise, poor growth, and poor appetite due to hypogeusia. However, zinc deficiency may be difficult to diagnose, as plasma zinc levels can be normal with true zinc deficiency. In patients with failure to thrive, six months of empiric zinc supplementation may be reasonable. Additionally, zinc can affect vitamin A, so it may be beneficial to supplement zinc in vitamin A deficiency settings that do not respond to vitamin A supplementation alone [3].

### 14.6. Essential Fatty Acids

Essential fatty acids are polyunsaturated fats that can be eventually metabolized from arachidonic acid (AA) into docosahexaenoic acid (DHA). This deficiency can be common in patients with CF, though it is not often symptomatic. Essential fatty acid deficiency should be considered in infants with failure to thrive. It is possible that in addition to fat malabsorption, patients with CF may have abnormal fatty acid metabolism. It is still unclear whether patients with CF should have supplementation with DHA, though foods rich in linolenic acid, such as cold-water fish and vegetable oils, can be sometimes recommended. Additionally, breastmilk contains DHA and may be beneficial for infants with CF [3,130].

## 15. Nutritional Impact of CF Therapy in the Era of HEMs

Patients with CF generally have lower body weight than age-matched peers. This may be due to increased energy expenditure secondary to increased work of breathing that leads to difficulty maintaining goal BMI. Improvements in CF management have resulted in overall improved nutritional status. In fact, there are now four times as many patients with CF who are overweight or obese compared to underweight [137].

Research suggests that CFTR modulators, such as elexacaftor/tezacaftor/ivacaftor, may increase weight in CF patients [138,139]. The mechanism behind this weight gain is thought to be multifactorial. Therapy can improve appetite, leading to increased food intake [12]. Improved CFTR function can lead to thinned mucus secretions and better nutrient absorption. Improved overall health may be correlated with fewer hospitalizations in which patients are required to fast. Additionally, elexacaftor/tezacaftor/ivacaftor therapy must be taken with dietary fat, which itself can increase calorie intake. Patients on adequate therapy may have improved respiratory function, which may correlate with decreased energy expenditure due to decreased respiratory muscle work. Therapy has also been associated with improved pancreatic exocrine function, which will theoretically improve intestinal nutrient absorption and thus nutritional status.

CFTR modulators have also been associated with increased high-density lipoprotein (HDL), low-density lipoprotein (LDL), and total cholesterol levels. This may be secondary to reduced systemic inflammation, as triglycerides are an acute-phase reactant. Inflammation and oxidative stress are linked to impaired glucose homeostasis, which may then theoretically improve with the use of CFTR modulators. Additionally, increased insulin secretion has been reported with CFTR modulator therapy [126].

Overall, data in adults suggest that the use of elexacaftor/tezacaftor/ivacaftor will increase the number of individuals who are overweight. Traditionally, nutrition recommendations for patients with CF who were generally undernourished and underweight in the past included high-fat and high-calorie diets. The Academy of Nutrition and Dietetics has altered their nutrition recommendations for patients with CF on HEM to reflect recommendations for the general population [137]. In conclusion, providers should monitor patients on CFTR modulator therapy for signs of overnutrition at every visit.

## 16. Conclusions

CF is a multisystemic chronic disease with many comorbidities playing a major role at the same time. This contributes significantly to existing nutritional challenges in this patient group and impacts nutritional management. CF providers have started seeing nutritional issues that are not considered typical in these patients, specifically, obesity. Multidisciplinary care from various providers (such as pulmonologists, gastroenterologists, nutritionists, otolaryngologists, endocrinologists) familiar with these issues is paramount to help manage the complex nutritional needs of patients with CF. CF is one of the many chronic diseases where the idea of co-production is truly needed, where providers, families, and patients all work interdependently to come together to establish common goals of care to improve care for CF patients.

## Figures and Tables

**Table 1 nutrients-14-01028-t001:** Comorbidities affecting nutrition in CF.

HEENT	Sinusitis
Respiratory	Chronic Lung Disease, Nocturnal Hypoxia
GI/Liver	Gastroesophageal reflux disease, gastroparesis, small intestine bacterial overgrowth, distal intestinal obstruction syndrome, constipation, pancreatitis, exocrine pancreas deficiency, enteropathies (e.g., celiac disease), cystic-fibrosis-related liver disease
Endocrine	Cystic-fibrosis-related diabetes
Psychosomatic	Disorders of eating, avoidant restrictive food intake disorder, disorders of gut–brain interaction
Micronutrients	Zinc deficiency, essential fatty acid deficiency, vitamin A, D, E, and iron deficiencies
HEM related	Overeating/obesity

## Data Availability

Not applicable.

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
