# Peer review of "Understanding Cystic Fibrosis Comorbidities and Their Impact on Nutritional Management"

_nutrients, 2022, doi:10.3390/nu14051028_

Round 1

Reviewer 1 Report

The authors write an informative and descriptive review on cystic fibrosis, its comorbidities and its nutritional implications.

  • Table 1: define acronyms (should be readable standalone)
  • Additional table: much of the important information could all be neatly summarized in one table
  • Lung disease:
    • 66: “causative relationship is not yet understood” – the following paragraph is written in a way that appears to show that poor nutrition leads to poor lung fxn. The quoted part can be rewritten to reflect this point
  • GERD
    • 119-123: Why/how does GERD exactly impact nutritional intake/status? (difficult to consume certain foods; emesis results in purging of nutrients, etc)
    • Recommend describing nutritional remedies of GERD. For e.g., avoid bothersome foods like fried food, mint, chocolate, caffeine, alcohol, spicy, acidic food; eat smaller, more frequent meals, keep food-symptom log.
  • GP
    • 158: is it 4 of the 9 patients with CF also had GP? If so, rephrase to make it clearer (9 out of 239 patients enrolled had CF, 4 of which had GP)
    • 159: what is the prevalence of GP in those without CF (to compare against those with CF)
  • SIBO
    • 225-232: describe how these malabsorptions are likely to affect the patient (i.e. clinical impact)
    • 239: how can amelioration of nutritional deficiencies be achieved, especially in a population struggling to consume adequately in the first place?
  • Constipation
    • 272: sentence is awkwardly written (“only limited studies” does not fit with the tone, suggest rephrasing
    • 276: no difference in what, exactly? Constipation symptoms?
    • 285: describe dietary sources of fiber. Also, exercise is another common and effective treatment
  • Other GI Enteropathies
    • 300: make the format of ratios consistent (either “1 in x” or “1:x”)
    • The presence of celiac disease further complicates nutritional intake for those with CF because it limits a common food group, one that contributes significantly to many people’s caloric intake. This added complication should be included in this section since those on a gluten-free diet will very likely encounter this as an obstacle.

Minor

  • PwCF is awkward acronym, suggest removing
  • Needs careful editing for syntax/grammar, capitalization (letter vitamins are not all capitalized (470), for example), spelling.
  • Oral supplements are often a mainstay of caloric supplementation in those with CF, especially those with poor appetite. There is room to expand on this in the review.
  • Abbreviations: ROME IV, ESPGHAN, CFRD, PERT, MI (some of these are not defined or used before being defined)
  • CF causes increased energy expenditure d/t difficulty breathing; increases energy requirements d/t mucus preventing nutrient absorption. This is explained late in the review, but should be explained in the beginning.

Reviewer 2 Report

The review “Understanding Cystic Fibrosis Comorbidities and Their Impact 2 on Nutritional Management” is a well written and extensive summary of nutritional challenges in people with CF. The new CFTR modulators are highly effective at improving CF disease, but also cause new problems as historically, CF patients have been underweight but now start to present with obesity. The review does not provide a solution, but is a summary of nutritional challenges faced by people with CF.

There are some concerns with the text:

Concerning overall consistency:

If you introduce the acronym CF for cystic fibrosis, you should use it throughout and be consistent.

Sometimes you write only the surname and sometimes you include initials. This should at least be consistent, if the journal has no guidelines.

Page 1, line 24-25: Inadequate functioning cystic fibrosis transmembrane conductance regulator (CFTR) protein caused by mutations in the CFTR gene disrupts the transport of sodium and chloride ions across cell membranes. Please explain how malfunctioning CFTR may disrupt sodium transport.

Table 1: Please explain the acronyms.

Page 2, line 41: people with cystic fibrosis (PwCF): You have already introduced that acronym, you do not introduce it again.

Page 3, line 76: Zemel et al, line 92: Welsh et al: There should be a full stop after al.

Page 3, line 84: Sino-pulmonary symptoms of cystic fibrosis: You have already introduced the acronym CF for cystic fibrosis. The acronym should then be used throughout.

Page 6, line 263: What do you mean by: associated with a history of MI? What is MI?

Page 7, line 294: Celiac disease (CD). The acronym CD is often used for Crohn´s disease, an inflammatory bowel disease. There is no need to have an acronym for celiac disease.

Page7, line 299: You write “compared to the healthy population”. I assume what you mean is “people without CF”. You cannot define them/us as healthy.

Page 7, line 300: 1 in 2,000,000 - 1:5,900,000: This expression should be consistent. Write either 1 in 5,900,000 or 1:2,000,000.

Page 7, line 303: standard

Page 8, line 352: their healthy peers. This assumes that if you do not have CF, you are healthy. It would be more correct to instead call them people without CF.

Page 9, line 389: cystic fibrosis transmembrane regulator (CFTR). You have already introduced that acronym. It should not be explained again.

Page 9, line 425: Was it really healthy peers or just people without CF?

Page 10, line 470: Capital A

Page 11, line 496: iron panel iron panel

Page 11, line 528: must been taken. Must be taken
